



# Evaluation of Global Teleconnections in CMIP6 Climate Projections using Complex Networks

Clementine Dalelane[1], Kristina Winderlich[1], and Andreas Walter[1]

[1]Deutscher Wetterdienst Frankfurter Str. 135 6305 Offenbach Germany

**Correspondence:** Clementine Dalelane (clementin.dalelane@dwd.de)

**Abstract.** In climatological research, the evaluation of climate models is one of the central research subjects. As an expression of large-scale dynamical processes, global teleconnections play a major role in interannual to decadal climate variability. Their realistic representation is an indispensable requirement for the simulation of climate change, both natural and anthropogenic. Therefore, the evaluation of global teleconnections is of utmost importance when assessing the physical plausibility of climate

projections.

We present an application of the graph-theoretical analysis tool $\delta$-MAPS, which constructs complex networks on the basis of spatio-temporal gridded data sets, here sea surface temperature and geopotential height in 500 hPa. Complex networks complement more traditional methods in the analysis of climate variability, like the classification of circulation regimes or empirical orthogonal functions, assuming a new non-linear perspective.

In the first place, $\delta$-MAPS assembles grid cells with highly coherent temporal evolution into so-called domains. In a second step, the teleconnections between the domains are inferred by means of the non-linear distance correlation. We construct two unipartite and a bipartite network for 22 historical CMIP6 climate projections and two century-long coupled reanalyses (CERA-20C and 20CRv3). The networks derived from projection data are compared to those from reanalyses using a similarity criterion borrowed from image processing.

## 1 Introduction

The evaluation of general circulation models (GCM) is one of the key topics of climate sciences. This evaluation is indispensable in the assessment of uncertainties in the projection of climate change. At the same time, it serves as a guideline for further model development.

Established methods of climate model evaluation include comparison of spatial and temporal means, and often also the

variability, of important climate parameters such as air temperature, precipitation, wind speed, geopotential height, radiation, and energy fluxes between model output and observational/reanalysis data (Zhang et al., 2021). More elaborated evaluation techniques assess the temporal evolution of global mean/sea surface/hemispheric temperature (Papalexiou et al., 2020) with respect to increasing greenhouse gas concentration or regional trends (Duan et al., 2021).

Acknowledging its importance for consistent climate simulation, Simpson et al. (2020) evaluate the atmospheric circulation

in terms of mean atmospheric fields, in combination with dynamical features like the jet stream, stationary waves and blocking.





In contrast, Kristóf et al. (2020) evaluated the positions of potential action centers of atmospheric teleconnections as a proxy for circulation.

Another approach is taken by Brands (2022) and Cannon (2020), who both assess circulation biases in correspondence to the representation of circulation types. Whereas Brands (2022) uses Lamb weather types, the analysis in Cannon (2020) is based on Principal Component Analysis (PCA) derived modes of variability. Such modes of variability, extracted by eigentechniques from spatio-temporal gridded data, have been the objective of evaluation efforts in recent years as their spatial patterns are supposed to reflect large scale dynamical processes in the climate system. For example, Fasullo et al. (2020) and Coburn and Pryor (2021) have assessed the representation of six oceanic and atmospheric modes in terms of spatial and spectral accuracy, including an evaluation of the interaction between modes. Still, it has been recognized that eigenmethods suffer from a number of limitations, because geometric constraints as linearity and normality, orthogonality and simultaneity do not correspond to physical properties of the climate system (Monahan et al., 2009; Fulton and Hegerl, 2021; Hynčica and Huth, 2020; Lee et al., 2019) and hinder their interpretation.

Besides, the evaluation of climate modes, such as El Niõ Southern Oscillation (ENSO) or North Atlantic Oscillation (NAO), is usually done at the component level. But it is the coupling among those components, which defines the large scale variability of climate at interannual and decadal time scales (Tsonis et al., 2008; Steinhäuser and Tsonis, 2014).

Complex network methods are able to account for non-linear, time-lagged, and high-order interactions in high-dimensional data, and have been introduced in climate sciences by the beginning of the 21$^{st}$ century (for an overview see Dijkstra et al. (2019)). Such networks investigate the interdependencies between all their constituent components, thereby unveiling dynamical features that could remain hidden to traditional analysis techniques. A rather fundamental property of climate networks is their organization in terms of communities—clusters of strongly connected nodes forming semi-autonomous subcomponents of the climate system with non-accidental similarity to many known modes of variability (Steinhäuser et al., 2009; Tsonis et al., 2011), that interact dynamically in multiple ways. Such emergent property has been ascribed to the mismatch between spatial and temporal scales on a sphere, which allows only a finite number of degrees of freedom (Yang et al., 2021).

The comparison of such complex network-derived communities between climate simulations and observation/reanalysis data sets was used for evaluation purposes first by Steinhäuser and Tsonis (2014). They assessed the community structure in climatic fields finding rather low consistency between the model runs and the reference data set. Likewise, Fountalis et al. (2015) assessed the community structure of model simulations, but complemented it with an evaluation of the interaction strength of the communities with ENSO. The idea was further developed by Fountalis et al. (2018) and Falasca et al. (2019) in their so-called $\delta$-MAPS approach to comprise a whole network of all communities, which is evaluated with regards to the distribution and size of communities, the interaction strength and the distribution of the links.

Note that there is another line of research into the evaluation of causal networks (for instance Vázquez-Patiño et al. (2019) or Nowack et al. (2020)), which is somewhat different to the approach followed here.

In the present article, we will explain (Sect. 3) and apply (Sect. 4) $\delta$-MAPS (Fountalis et al., 2018) to construct functional networks for sea surface temperature (SST) and geopotential height in 500 hPa (Z500) fields, as well as a cross-network between SST and Z500, using GCM output data from the Coupled Model Intercomparison Project Phase 6 (CMIP6). We



compare the derived networks to analogous networks from reanalysis data, namely CERA-20C (Laloyaux et al., 2018) and 20CRv3 (Slivinski et al., 2019), to evaluate the capacity of the GCMs in reproducing complex non-linear processes in the atmosphere and the ocean.

This assessment is all the more instructive as it is not possible to tune the teleconnections directly. In nature and in models, teleconnections emerge from the interplay of the governing equations under the condition of the boundaries. A models gets them right, if and only if the model specifications are sufficiently well approximated and well balanced between model components.

## 2 Data

The objective of the present study is to compare the interaction networks derived from CMIP6 GCM output from historical simulations to reference networks derived from *two* century-long reanalyses in order to account for uncertainties in observations and differences in construction methods as recommended by Hynčica and Huth (2020); Lee et al. (2019) and others: (i) the Coupled Reanalysis for the 20$^{\text{th}}$ Century (CERA-20C) provided by the ECMWF (Laloyaux et al., 2018) (10 ensemble members and ensemble mean), and (ii) from the NOAA-CIRES-DOE Twentieth Century Reanalysis version 3 provided by the NOAA/OAR/ESRL PSL (Slivinski et al., 2019) (best estimate).

The presented study is intended to help the selection of physically plausible GCM runs for further dynamical downscaling in the Coordinated Downscaling Experiment–European Domain (https://www.euro-cordex.net/). Therefore, the CMIP6 model ensemble evaluated here follows the list of model runs under consideration in EURO-CORDEX, for which all necessary forcing data had been provided at the time of writing, plus some extra models (Table 1).

We consider the parameters sea surface temperature (SST) and geopotential height at 500 hPa (Z500). These are relatively well-observed and smoothly varying fields suitable for the construction of networks. Steinhäuser et al. (2012) confirm good network properties for SST and Z500 with many proximity-based correlation links as well as a large number of teleconnections. In accordance, Donges et al. (2011) found the maximal link density for geopotential in about 4 to 6 km height, and Wiedermann et al. (2017) detected the highest transitivity between SST and geopotential height in 500–300 hPa.

From the coupled network perspective, it would be highly desirable to include further parameters into the analysis like sea surface salinity, or, still more interesting, variables from the stratosphere and the deep ocean. Unfortunately, the observations of such parameters are only recently becoming more reliable and less sparse, such that the fidelity of their reanalyses fields is impossible to verify.

The SST (Z500) data was remapped to a common grid of $2.25° \times 2.25°$ ($2.5° \times 2.5°$) resolution. Regions with sea ice are avoided in SST as well as circles of $5°$ radius around the poles in Z500 because of possibly biased representation of the polar vortices. The analysis is carried out for seasonal anomalies on the overlapping time period from 1901 to 2010.



**Table 1.** CMIP6 models

| Model | hist. exp. | Reference |
|---|---|---|
| ACCESS-CM2 | r1i1p1f1 | Bi et al. (2020) |
| ACCESS-ESM1-5 | r1i1p1f1 | Ziehn et al. (2020) |
| BCC-CSM2-MR | r1i1p1f1 | Wu et al. (2019) |
| CanESM5 | r1i1p2f1 | Swart et al. (2019) |
| CESM2 | r2i1p1f1 | Danabasoglu et al. (2020) |
| CMCC-CM2-SR5 | r1i1p1f1 | Cherchi et al. (2019) |
| CMCC-ESM2 | r1i1p1f1 | Cherchi et al. (2019) |
| CNRM-CM6-1 | r1i1p1f2 | Voldoire et al. (2019) |
| CNRM-ESM2-1 | r1i1p1f2 | Séférian et al. (2019) |
| EC-Earth3 | r1i1p1f1 | Döscher et al. (2022) |
| EC-Earth3-Veg | r1i1p1f1 | Döscher et al. (2022) |
| HadGEM3-GC31-LL | r1i1p1f3 | Roberts et al. (2019) |
| IPSL-CM6A-LR | r1i1p1f1 | Boucher et al. (2020) |
| MIROC6 | r1i1p1f1 | Tatebe et al. (2019) |
| MIROC-ES2L | r1i1p1f2 | Hajima et al. (2020) |
| MPI-ESM1-2-LR | r1i1p1f1 | Gutjahr et al. (2019) |
| MPI-ESM1-2-HR | r1i1p1f1 | Müller et al. (2018) |
| MRI-ESM2-0 | r1i1p1f1 | Yukimoto et al. (2019) |
| NorESM2-LM | r1i1p1f1 | Seland et al. (2020) |
| NorESM2-MM | r1i1p1f1 | Seland et al. (2020) |
| TaiESM1 | r1i1p1f1 | Lee et al. (2020) |
| UKESM1-0-LL | r1i1p1f2 | Sellar et al. (2019) |

## 3 Methods

The procedure used to assign an assessment score to each model run comprises a number of algorithmic stages that build on each other. As they are not yet well-known in the climatological community, we present them in detail in the following subsections:

– Detrending with trend-EOF (Sect. 3.1)

    – Network construction with $\delta$-MAPS (Sect. 3.2)

        – Domain identification (Sect. 3.2.1)

        – Network of domains (Sect. 3.2.2)

    – Distance covariance and distance correlation (Sect. 3.3)



– Distance multivariance and distance multicorrelation (Sect. 3.3.1)

     – Network comparison with structural similarity index and multivariate network quality score (Sect. 3.4)

## 3.1 Detrending with trend-EOF

Prior to the construction of the $\delta$-MAPS networks, the data has to be detrended to avoid the correlations being distorted by long-term trends. Although it is still the most widely used technique, linear detrending has been shown little appropriate to remove

the effects of external forcing (anthropogenic and natural) from climatic time series (Frankignoul et al., 2017), given its non-linear structure and the dynamical response mechanisms including long-range memory. Conventional Empirical Orthogonal Function (EOF) decomposition is not well suited for trend detection either for a number of reasons (Hannachi, 2007), which often cause the spreading of long-term trends between several modes of internal variability. Instead, we will apply a non-parametric technique, so-called trend-EOF (Hannachi, 2007), which identifies spatial patterns of trends defined as common

non-linear, but monotone increase. The method is based on the Singular Value Decomposition (SVD) of the matrix of inverse ranks, instead of the direct observations as in conventional EOF-analysis. Since sequences of inverse ranks provide a robust measure of monotonicity, trend-EOFs are able to separate patterns associated with monotone (non-linear) trends, albeit small, from patterns not associated with trends.

     Trend-EOFs have been applied since in a number of studies (e.g. Barbosa and Andersen (2009), Li et al. (2011), Meegan Ku-

mar et al. (2021) among others). Fisher (2015) compared trend-EOFs, along with conventional EOFs, to a selection of other PCA-based techniques, which are designed to extract space-time patterns maximizing criteria like persistence, predictability or autocorrelation. In contrast to conventional EOFs, all the tested methods very robustly detect a leading EOF pattern with a respective principal component (PC) that presents a distinct non-linearly increasing trend. We consider trend-EOFs therefore an appropriate technique for identifying anthropogenic Greenhouse Gas (GHG)-forced trends.

Let $\mathbf{X} = ((x_{it}))$ be the matrix of anomaly data at grid cells $i = 1 \ldots n$ (numbered consecutively) and times $t = 1 \ldots T$. The time series $\boldsymbol{x}_i$ at grid cell $i$ is transformed to the vector of inverse ranks $\boldsymbol{q}_i$ by setting $q_{it}$ equal to the time position of $t^{\text{th}}$ largest value in $\boldsymbol{x}_i$. The sequence $\boldsymbol{q}_i$ indeed reflects the total monotonicity of $\boldsymbol{x}_i$: in monotone series the inverse ranks are ordered according to the trend. The stronger the trend in $\boldsymbol{x}_i$, the stronger the pattern in $\boldsymbol{q}_i$. By maximizing the correlation in $\mathbf{Q} = ((q_{it}))$, we find a common trend that is shared (to some extent) by all grid cells, which makes sense in the light of

GHG-forced warming.

     After centering and cosine weighting of $\mathbf{Q}$ w.r.t. the corresponding latitude, the principal components and the loading patterns are obtained by SDV: $\mathbf{Q} = \mathbf{U}\boldsymbol{\Sigma}\mathbf{V}^{\mathbf{T}}$. The trend is now concentrated in the first (few) principal component(s), strongly distinguished by high eigenvalue(s) outstanding over the remaining low and slowly descending spectrum. If second or third order outstanding eigenvalues should be detected, they indicate additional, regionally confined independent trends, which are

generated by internal dynamical feedback processes. For our purpose of identifying regions with coherent time evolution, we would therefore want to retain such regional trends and eliminate only the trend associated to the first trend-PC. Likewise, regional trends caused by volcanic eruption are most probably not filtered either by the first trend-EOF. However, the impacts





of 20[th] century eruptions lasted only for short time periods, and on the other hand they are not well represented in surface-input reanalyses like CERA-20C and 20CRv3 (Fujiwara et al., 2015). We therefore assume that our evaluations remain valid. The

first trend-PC $u_1$ is now transformed back to physical space by projection $w_1 = \mathbf{X}u_1$, and the corresponding spatial pattern is composed of the regression coefficients between the trend-PC $w_1$ and the anomaly time series of the original field $x_i$.

To allow for an annual cycle in the trend patterns, we extend the trend-EOFs in analogy to season-reliant EOFs (Wang and An (2005), see also cyclo-stationary EOFs in Yeo et al. (2017)), $\mathbf{Q} = (\mathbf{Q}_{MAM}|\mathbf{Q}_{JJA}|\mathbf{Q}_{SON}|\mathbf{Q}_{DJF})$ (seasonally centered, inverse ranks calculated for each season individually), which extract a recurrent sequence of seasonal trend patterns with one

associated trend-PC for the magnitude of the whole cycle as opposed to one common pattern for all seasons as in non-seasonal EOF analysis or four individual patterns with their associated individual PCs as in seasonal EOFs, respectively. At this stage it would be possible to apply a secondary SVD to the seasonal warming patterns to obtain a smoother annual cycle. While such procedure seems undue for seasonal data, it would be a reasonable approach in the case of monthly data. Instead of applying two sequential EOFs to $\mathbf{Q}$, a tensor decomposition like HOSVD (De Lathauwer et al., 2000) would serve this purpose more

elegantly.

After having detrended the time series, we are able to standardize the seasonal variances without the interference of the seasonal trends, which would otherwise bias our estimates. On their part, seasonally varying variances could degrade the estimated correlations between grid cells in the first stage of the $\delta$-MAPS algorithm, giving increased weight to seasons with higher variance. In turn, the *spatial* component of the variance will be important in the second stage of $\delta$-MAPS, therefore we

augment the deseasonalized time series again with their overall (non-seasonal) variance.

## 3.2 Network construction with $\delta$-MAPS

### 3.2.1 Domain identification

To infer the functional interactions within and between spatio-temporal gridded datasets of climatological parameters, we adopt the $\delta$-MAPS algorithm proposed by Fountalis et al. (2018). This algorithm is rooted in network sciences/graphical modelling,

in which graphs are used to express the dependence structure between random variables. A graph or network consists of a set of nodes connected by a set of edges, which describe the interactions between the nodes. Networks can be classified depending on their topology: simple networks like lattices and fully-connected networks; complex networks like scale-free and small-world networks. Small-world networks are often observed in climate and other earth sciences, in the human brain and in social networks. Their nodes are strongly clustered into semi-autonomous components and the average shortest path length between

any two nodes is small.

In contrast to structural networks or flow networks, where the edges are physically observable (like wired connections or trajectories of particles, respectively), functional networks are inferred from the behaviour of the nodes. We consider the grid cells of a selected climatological field as the nodes of the graph. The spatial embedding is naturally given by the locations of the grid cells. In Fountalis et al. (2018) the edges of a fully-connected grid cell-level network are defined using the unpruned

Pearson correlation $\varrho$ of the time series as association measure between any pair of nodes. Based on this weighted network,



the $\delta$-MAPS algorithm identifies semi-autonomous components $D_1 \ldots D_K$, called domains. A domain is a spatially contiguous set of grid cells with highly correlated temporal activity. Fountalis et al. (2018) propose an iterative algorithm that alternately expands and merges a preliminary set of domain-seeds $S$ (neighborhoods with locally maximal correlation, $3 \times 3$ grid cells in our case) so as to find the maximum possible sets of grid cells that satisfy the homogeneity constraint $\delta$: Let $D$ be a spatially

contiguous set of grid cells with cardinality $|D|$

$$\delta \quad \leq \quad \varrho_D \quad := \quad \frac{1}{|D|(|D|-1)} \sum_{i \neq j \in D} \varrho_{ij} \tag{1}$$

where $\varrho_{ij}$ is the Pearson correlation between the time series at grid cells $i$ and $j$ and $\delta$ is a chosen parameter to regulate the number and size of the domains. The domains are expanded to neighboring grid cells (one at a time) as long as $\varrho_D \leq \delta$. Two domains $D_i$ and $D_j$ are merged if they contain at least one pair of adjacent grid cells and their union still satisfies the threshold

$\delta$. The algorithm stops, when no more domains can be merged or expanded.

The number of domains $K$ generated by this algorithm is not predefined. Overlapping domains are allowed in $\delta$-MAPS, because grid cells might be influenced by more than one physical process. If a grid cell does not satisfy the homogeneity constraint with any of its neighbors, it remains unassigned. Deviating from Fountalis et al. (2018), we use Spearman's Rank correlation to determine the similarity between grid cells to allow for monotone, yet non-linear association. Furthermore, we

set the threshold $\delta$ for minimal average correlation within a domain to equal a selected high quantile of all pairwise correlations (our $\delta$ is not based on a significance test, therefore there is no need to correct for auto-correlation). Lower thresholds allow the domains to expand and merge further resulting in a smaller number of spatially larger domains, which means lower parcellation, and vice versa. In Sect. 4, we will choose $\delta$ so as to produce "intuitive" domains evocative of known teleconnection patterns.

In Falasca et al. (2020), the identification of domains was further refined: grid cells are assigned to a common domain

if their time varying complexity (quantified by recurrence entropy) evolves coherently. Coherent evolution of complexity reflects coherent dynamical evolution and is thus an even stronger indicator for semi-autonomous component organisation than correlation between the original climatological time series. But for complexity time series to construct, the proposed recurrence measure has to be evaluated on moving time windows (100 year windows over 6000 years of monthly values in Falasca et al. (2020)). Unfortunately, our time series are not long enough to detect complexity changes by means of recurrence entropy (nor

to actually occur in the real climatological fields), so we have to stick to the original definition of $\delta$-MAPS in Fountalis et al. (2018).

The first stage of $\delta$-MAPS is a local community detection algorithm, where the criterion to maximize is the number of grid cells assigned to a minimum number of communities under the conditions (i) $\varrho_D \geq \delta$, (ii) $D$ is spatially contiguous, and (iii) $D$ contains a seed $s \in S$ (Fortunato and Hric, 2016). As this problem is NP-hard (solvable in polynomial time, Fountalis et al.

(2018)), the greedy algorithm of Fountalis et al. (2018) only approximates one possible solution. Even though, it is able to detect meaningful communities of any size (no preferred scale) and independently from the network structure in other spatial regions.



### 3.2.2 Network of domains

Subsequently, the domains identified above serve as super-nodes in the second stage of $\delta$-MAPS. A functional weighted
network is inferred between the domains on the basis of a dependence measure (in Fountalis et al. (2018) the lagged moment
correlation is used; we will use distance correlation, see Sect. 3.3). The time series of a domain is defined as

$$(x_{D1}\dots x_{DT}), \quad x_{Dt} = \frac{1}{\sum_{i\in D}\cos\varphi_i}\sum_{i\in D}x_{it}\cos\varphi_i \tag{2}$$

where $\varphi_i$ is the latitude of grid cell $i$. In contrast to Falasca et al. (2019), we use the means instead of the sums of the grid
cells for domain time series. We do so because otherwise the variances of the domains would grow with their size, something
that would hinder interpretation. On the other hand, the spatial correlation within the domains, the precondition for grid cells
to form a domain, impedes the decrease of the variance of the domain mean following the Central Limit Theorem at the rate of
$\sqrt{|D|}$. Instead, the variances of the domain means are of comparable magnitude regardless of the domain size.

Every possible link with every possible lag $-L \leq l \leq L$ is tested for significance, which constitutes a multiple-testing prob-
lem such that the cumulative probability of type I errors increases. One way to control the false discovery rate FDR to be
smaller than a predefined level $\alpha$ was proposed by Benjamini (2010): the $p$-levels of the individual tests are ordered ascend-
ingly, $p_{(1)} \leq \cdots \leq p_{(\frac{1}{2}K(K-1)(2L+1))}$, and the hypothesis ($H_0$: link is insignificant) is rejected only for those tests, where
$p_{(k)} < \frac{2k\alpha}{K(K-1)(2L+1)}$.

The network consists of two maps $D$ and $W$. $D$ : set of nodes (grid cells) $\longrightarrow$ power set of domains $\mathcal{P}\left(D_1\dots D_K\right)$, which
assigns one/several/no domains to every grid cell, and $W$ : set of pairs of domains $\{D_1\dots D_K\} \times \{D_1\dots D_K\} \longrightarrow$ maximal
(lagged) dependence $\in \mathbb{R}$, which assigns every pair of domains a link that equals the maximal (lagged) dependency between
them (we allow lags up to 10 seasons).

The distinction between grid cells that are dependent within the same domain and grid cells that are dependent across two
different domains allows $\delta$-MAPS to differentiate between local diffusion phenomena and remote interactions as for instance
an atmospheric bridge or an oceanic tunnel (Liu and Alexander, 2007).

Since the techniques to construct the $\delta$-MAPS network are statistical, long time series are convenient in order to obtain
robust estimates of the dependence measures. In the case of non-stationarity, such estimates would be biased and reflect only a
temporal average connectivity between the components of the network. The time dependence can be addressed using evolving
networks, which are constructed over sliding time windows (see for instance Kittel et al. (2021) and Novi et al. (2021)). The
present study considers a time-constant network for the period 1901-2010, and a shorter period network for 1951-2010, where
more observations are available for assimilation into the reanalyses. To investigate the temporal evolution, a third network is
constructed for 1901-1955.

The complex networks framework offers a lot more approaches in order to exploit the richness of the data, as for instance
multi-scale, causal, and multi-layer networks. Wavelet multi-scale networks were proposed for investigating interactions in the
climate system simultaneously at different temporal scales, revealing features which usually remain hidden when looking at
one particular time scale only (Agarwal et al., 2018, 2019). Interactions between processes evolving on different time scales are



investigated by Jajcay et al. (2018). Moreover, as the number of identified domains within a climatological field is drastically smaller than the number of original grid cells, this also opens up the possibility of investigating the causal relationships between them (Nowack et al., 2020), although the basic assumption of causal networks inference, that the dependence structure can be represented by a directed acyclic graph, is questionable in the climate context. The construction of both dependence based and

causal networks can naturally be extended to cross-networks, which include multiple fields (Feng et al., 2012; Ekhtiari et al., 2021).

### 3.3 Distance covariance and distance correlation

As physical processes in climate are highly dynamical and mostly non-linear (Donges et al., 2009), we decided to substitute the Pearson correlation in the second step of network inference by a non-linear dependence measure: distance correlation

proposed by Székely et al. (2007). To begin with, distance covariance, calculated from the pairwise Euclidean distances within each sample, is an analogue to the product-moment covariance, but it is zero if and only if the random vectors are independent. The intuition of distance covariance is that if there exists a dependence between the random variables $X$ and $Y$, then for two similar realizations of $X$, say $x_s$ and $x_t$, the two corresponding realizations of $Y$, $y_s$ and $y_t$, should be similar as well. Note that the opposite ($x_s$, $x_t$ unsimilar $\implies y_s$, $y_t$ unsimilar) is true for linear dependence, but not true in general.

Unlike the widely used information measures, distance covariance has a compact representation, is computationally fast, and reliable in a statistical sense for sample sizes common in climatology, because it is not necessary to estimate the density of the samples. We use the unbiased version of distance covariance given in Székely and Rizzo (2014). Let $(x_t), (y_t), t = 1 \ldots T$ be a statistical sample from a pair of real or vector valued random variables $X$, $Y$. First, compute all pairwise Euclidean distances:

$$a_{st} = \|x_s - x_t\|_2 \quad \text{and} \quad b_{st} = \|y_s - y_t\|_2,$$

and perform a double centering for all $s \neq t$

$$A_{st} = a_{st} - \frac{1}{T-1}\sum_u a_{su} - \frac{1}{T-1}\sum_v a_{vt} + \frac{1}{(T-1)(T-2)}\sum_{uv} a_{uv}$$
$$B_{st} = b_{st} - \frac{1}{T-1}\sum_u b_{su} - \frac{1}{T-1}\sum_v b_{vt} + \frac{1}{(T-1)(T-2)}\sum_{uv} b_{uv}.$$

Then distance covariance dCov is definded as

$$\mathrm{dCov}(X,Y) := \frac{1}{T(T-3)}\sum_{st} A_{st}B_{st} \tag{3}$$

Distance variance dVar and distance correlation dCor are defined analogously to moment variance and moment correlation, resepctively:

$$\mathrm{dVar}(X) = \mathrm{dCov}(X,X) \quad \text{and} \quad \mathrm{dCor}(X,Y) := \frac{\mathrm{dCov}(X,Y)}{\sqrt{\mathrm{dVar}(X)\mathrm{dVar}(Y)}} \tag{4}$$

Distance correlation has a number of desirable properties:





1. $0 \leq \mathrm{dCor}(X,Y) \leq 1$

2. $\mathrm{dCor}(X,Y) = 0 \iff X, Y$ independent

3. $\mathrm{dCor}(X,Y) = 1 \iff Y$ is a linear transformation of $X$

Distance correlation is furthermore robust against auto-dependence (K. Fokianos and M. Pitsillou, 2018), which eliminates the need to correct for autocorrelation, as it was done in Fountalis et al. (2018). The correction of autocorrelation is a rather unrobust statistical technique, meaning that it frequently yields spurious results, such that its expendebility is statistically advantageous.

An efficient test of distance correlation based on the $\chi^2$-distribution was proposed by Shen et al. (2022), which is universally consistent, and valid for $\alpha \leq 0.05$:

$$\Phi(X,Y) = \begin{cases} 1 & \text{if} \quad T\,\mathrm{dCor}(X,Y) \geq F^{-1}_{\chi_1^2-1}(1-\alpha) \\ 0 & \text{else} \end{cases} \tag{5}$$

Distance correlation is defined between vectors of arbitrary dimension. One way to take advantage of this property in the construction of networks would be to assign the measurement of more than one climatological variable to every node, e.g. sea surface temperature and salinity, or 500 hPA geopotential height and temperature.

We will apply distance correlation in the network inference between the domains, but not in the construction of the domains. The reason is that in domain construction we are looking for similar temporal behavior between grid cells. We choose Spearman's Rank correlation, because it accounts for non-linear, yet monotone association. In contrast, in network inference we are expressly interested in non-linear dependence including non-monotonicity.

### 3.3.1 Distance multivariance and distance multicorrelation

Distance correlation has also been generalized to distance multivariance/multicorrelation by Böttcher et al. (2019) to measure the dependence between an arbitrary number $n$ of random variables in the sense of Lancaster interaction (Lancaster, 1969; Streitberg, 1990). The Lancaster interaction $\Delta F$ quantifies the fraction of dependence between them that is not explained by factorization, their *synergy*. For $n = 3$, let $F_{123}$ be the 3-dimensional joint distribution function of $X_1, X_2, X_3$, $F_{12}$, $F_{13}$ and $F_{23}$ the pairwise joint and $F_1$, $F_2$ and $F_3$ the marginal distribution functions. Then the Lancaster interaction is defined as

$$\Delta F = F_{123} - F_1 F_{23} - F_2 F_{13} - F_3 F_{12} + 2 F_1 F_2 F_3$$

the fraction of $F_{123}$ that is not explained by pair-wise dependence. Lancaster interaction excludes, in particular, linear dependence as this is indeed explained by pair-wise dependence.

The concept of higher-order dependence is related to joint cumulants and higher-order moments, in that $\kappa_n(X_1 \ldots X_n) = \int x_1 \ldots x_n d\Delta F$ (Streitberg, 1990). Joint cumulants are traditionally applied in multiple-point statistics and hyper-spectral analysis to describe non-linear interaction and non-gaussian multidimensional distributions. Climate science has seen only a small number of implementations, including the contributions of C. A. L. Pires related to teleconnections (e.g. Pires and



Hannachi (2017, 2021)). As a feature of complex systems, higher-order interactions have already been recognized as critical
for the emergence of complex behavior such as synchronization and bifurcation in scientific fields as diverse as social networks
science, ecology, molecular biology, quantum physics, neurosciences, epidemics, geodesy, image processing and genetics
(Battiston et al., 2020), and tools for the construction of hypergraphs (graphs with links that comprise more than two nodes)
are increasingly available. To our knowledge, hypergraphs have not yet been introduced in climatology.

Distance multivariance is defined analogously to distance variance (Equation 3) and is a strongly consistent estimator of
Lancaster interaction (Böttcher et al., 2019). For $n = 3$, with $C_{st}$ the analogue to $A_{st}$ and $B_{st}$ for a third random variable $Z$:

$$\text{dMvar}(X, Y, Z) = \frac{1}{T(T-3)} \sum_{st} A_{st} B_{st} C_{st} \tag{6}$$

and likewise distance multicorrelation with a slightly differing normalisation:

$$\text{dVar}_3(X) := \text{dMvar}(X, X, X) \quad \text{and} \quad \text{dMcor}(X, Y, Z) = \frac{\text{dMvar}(X, Y, Z)}{\left(\text{dVar}_3(X) \cdot \text{dVar}_3(Y) \cdot \text{dVar}_3(Z)\right)^{1/3}} \tag{7}$$

Obviously, distance covariance between two random variables is covered by distance multivariance for $n = 2$. Significance
tests for distance multivariance are also given in Böttcher et al. (2019). As the asymptotic test is conservative and furthermore,
in the case of non-zero pairwise dependence, the test statistic is not guaranteed to diverge, it is convenient to choose a larger
FDR level than the usually employed significance levels between 0.1 and 0.01.

### 3.4 Comparison of networks with structural similarity index and multivariate network quality score

This study aims to compare the interaction networks derived from CMIP6 model output to the selected reference networks.
Our metric of comparison netSSIM is a modification of the netCorr criterion for functional networks developed by Falasca
et al. (2019). netCorr is a sophisticated metric, which evaluates the differences in topology and connectivity, combined in the
adjacency matrix $\mathbf{M}$ of each network, simultaneously. Let $\mathbf{M} = ((M_{ij}))_{i,j=1}^{n}$ be a square matrix of dimension $n$ (number of
grid cells) with

$$M_{ij} := \begin{cases} 0 & \text{if} \quad D(\boldsymbol{x}_i) = \emptyset \text{ or } D(\boldsymbol{x}_j) = \emptyset \\[2ex] \frac{1}{|\{W(D(\boldsymbol{x}_i), D(\boldsymbol{x}_j)) > 0\}|} \sum_{D_k \in D(\boldsymbol{x}_i), D_l \in D(\boldsymbol{x}_j)} W(D_k, D_l) & \text{if} \quad \emptyset \neq D(\boldsymbol{x}_i) \neq D(\boldsymbol{x}_j) \neq \emptyset \end{cases} \tag{8}$$

where $W(D_k, D_l) = \text{dCor}(\boldsymbol{x}_{D_k}, \boldsymbol{x}_{D_l})$. Alternatively, M could be rearranged in a 4-modal hypermatrix or tensor made of the
Kronecker product of the lat-lon field times itself containing the dependencies between the grid cells.

Apart from replacing Pearson by distance correlation, our definition of $\mathbf{M}$ differs from the one in Falasca et al. (2019) in three
aspects. Firstly, our links are undirectional, because distance correlation is much less sensitive to temporal lag than Pearson
correlation, such that there is no distinct temporal ordering. Secondly, we have defined $W(D_k, D_k) = 1$, causing $M_{ij} = 1$ if
$\boldsymbol{x}_i$ and $\boldsymbol{x}_j$ pertain to the same domain (and no other) to emphasize that grid cells within one domain are more strongly linked
to each other than to the grid cells of other domains. Thirdly, we set $M_{ij}$ the average of the links between domains that $\boldsymbol{x}_i$ and



$x_j$ belong to instead of the maximum as a means to account for overlapping domains. We do not apply any weighting to this average, because the mean internal rank correlation within each domain, i.e. the bond of a grid cell to its domains, is equally $\approx \delta$ by construction.

netCorr between two networks measures the spatial correlation between the respective adjacency matrices, not considering the overall level and variability within the networks. We propose to augment netCorr to netSSIM. SSIM is the Structural Similarity Index, a measure very popular in image processing, which combines terms for brightness (mean), contrast (variance) and structure (pattern correlation) of images (Wang et al., 2004). It was introduced to the hydrological/meteorological community by Mo et al. (2014). Let $X, Y$ be two gridded fields.

$$\text{SSIM}(X,Y) = \frac{2\mu_X\mu_Y + c_1}{\mu_X^2 + \mu_Y^2 + c_1} \cdot \frac{2\sigma_X\sigma_Y + c2}{\sigma_X^2 + \sigma_Y^2 + c2} \cdot \frac{\sigma_{XY} + c_3}{\sigma_X\sigma_Y + c_3} \tag{9}$$

where $\mu_X$, $\mu_Y$ are the means, $\sigma_X^2$, $\sigma_Y^2$ are the variances and $\sigma_{XY}$ is the Pearson covariance between $X$ and $Y$, and small constants $c_1, c_2, c_3$ to ensure regularity. The SSIM ranges from $-1$ to $1$, it equals $1$ only in case of identity and $-1$ for an anti-analogue (equal mean and variance, but correlation$=-1$). SSIM$= 0$ means no similarity. Note that the SSIM is not invariant under translation and rotation, which corresponds to our requirements, because we want the teleconnections to sit in the right
place. SSIM is not a distance metric, but a distance metric can be constructed from it (Brunet et al., 2011).

Falasca et al. (2019) recommend the use of their netCorr criterion always in combination with a criterion comparing the strength of the interaction, which they define as the sum of the links of a particular domain in terms of covariance. We argue that the strength is a criterion that intermingles the distribution of interactions between the domains with the variances of the domains, which, in turn, are determined by the size of the domains and the variances of the included nodes. We therefore prefer
to evaluate the interactions on their own using the netSSIM. The evaluation of the variances (or standard deviations) of model output data is a task that is already routinely performed in conventional evaluation setups.

We apply the (latitude weighted) SSIM to two adjacency matrices $\mathbf{M}$ (Equation 8) constructed from the significant distance correlations in two reference and/or model networks. In this way, we calculate netSSIM indices for the unipartite networks for SST and Z500, and for the cross-networks between the SST and Z500 domains.
Alternatively, we could calculate the SSIM between adjacency matrices in a point-wise manner, comparing the slices of the 4-modal hyper-matrices that correspond to the links of one individual grid cell to all others and then taking the weighted mean of all point-wise SSIMs.

Finally, we define a Network Quality Score (NQS) by applying an exponential transform to the netSSIMs, which projects them to the interval $[0,1]$ (recall that the netSSIM lives on $[-1,1]$). The same transform was used in Sanderson et al. (2015)
and Brunner et al. (2020) to construct quality scores from error measures, which are later fed into a model selection algorithm.

$$\text{NQS} := \exp\left\{-(1 - \text{netSSIM})^2\right\} \tag{10}$$

In order to combine the three NQSs wrt. SST, Z500 and SST–Z500, we take the geometric mean (equal to the exponential of the arithmetic mean of the squared differences $(1 - \text{netSSIM})^2$). This shall be the Multivariate Network Quality Score MNQS:



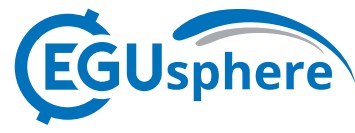


$$\text{MNQS} := (\text{NQS}_{\text{SST}} \cdot \text{NQS}_{\text{Z500}} \cdot \text{NQS}_{\text{SST-Z500}})^{\frac{1}{3}} \quad = \quad \exp\left\{-\frac{1}{3}\|\mathbf{1} - \text{netSSIM}\|^2\right\} \tag{11}$$

The MNQS corresponds to the exponential transform of the squared Euclidean distance between the 3-dimensional vector-netSSIM to the ideal vector-netSSIM value $(1, 1, 1)$, which would be attained by a network identical to the reference, normalized with the distance between $(1, 1, 1)$ and the value $(0, 0, 0)$ indicating no similarity.

Any other vector norm could be utilized for the construction of MNQS, for instance an $L_p$-norm with $p \neq 2$ or some weighting of the directions. netSSIMs for additional parameters can be incorporated into the MNQS in a straight forward way. Finally, the considered models can be ranked with respect to these scores.

    The netSSIM is also useful, when exploring the differences between networks in more detail. As mentioned above, the slices of $\mathbf{M}$ wrt. a single grid cell or domain can be compared one by one. It is further possible to calculate the netSSIM for
all pairwise links in a certain region, excluding the rest of the globe, or for all links from one region to another. This way, differences across models or time periods can be tracked down directly to their origin.

## 4   Results and discussion

We will demonstrate the functioning of every sub-procedure considering the CERA-20C ensemble mean over the whole period 1901–2010 as an example. All procedures are furthermore applied to the periods 1901–1955 and 1951–2010. Individual runs
of CERA-20C as well as 20CRv3 and CMIP6 model realisations will be discussed depending on special interest.

### 4.1   Detrending with trend-EOF

Trend-EOFs (Hannachi, 2007), as introduced in Sect. 3.1, produce time series of common change (SST and Z500) generated from the trend-PCs in the inverse-rank space, and the respective trend-loading patterns, (four seasonal trend-loading patterns per trend-PC in the case of season-reliant/cyclo-stationary trend-EOFs), indicating regions of stronger/weaker change. As
expected, the increase in SST is concentrated in the first trend-PC (the leading eigenvalues are 30 to 50 times higher than the trailing ones), the other trend-PCs showing no secular trend. Figure S1(a) depicts the global mean sea surface temperature anomaly (GMSSTa) (wrt. the reference period 1951–1990) in the CERA-20C ensemble mean, the forced temperature increase estimated by the first trend-EOF and the detrended anomalies. The grid cell-wise detrended anomalies are deseasonalized with regard to variance.

The GMSSTa derived from trend-EOFs in all runs of CERA-20C (not shown) as well as in the ensemble mean show a very similar evolution among each other and to Zhu et al. (2018), the breakpoints in temperature increase postulated therein at 1942, 1975 and 2004 clearly discernible. Likewise, the physical space-loading patterns of the ensemble mean (Figure S1 (b)) and all runs of CERA-20C are very similar to each other and and resemble the leading modes extracted using slow feature analysis and dynamical mode decomposition in Fulton and Hegerl (2021), identified as warming trends.



Analogous plots for geopotential height anomalies in 500 hPa for the CERA-20C ensemble mean can be found in Figure S2. Unfortunately, we were not able to find any comparable study in the literature, where Z500 was analysed for trend over the 20$^{th}$ century. Gillett et al. (2013), Knutson and Ploshay (2021), Garreaud et al. (2021), and Raible et al. (2005) considered SLP trends over different time periods and regions. Although not fully comparable, there is a certain similarity.

    The projected trends as well as the loading patterns in the 20CRv3 best estimate are somewhat different for the period 1901– 385  2010, but agree much better for 1951–2010 (not shown). This might well be related to low observational coverage during the first half of the century, we thus take this disagreement as a signal for caution.

    When subject to the same procedure, the CMIP6 model output SST and Z500 anomalies produce trend-EOFs and loading patterns roughly similar to CERA-20C and 20CRv3 (not shown). Differences are more or less obvious, though, such that an evaluation of the GMSST anomaly time series in the spirit of Papalexiou et al. (2020) would be an obvious choice, but is out 390  of the scope of this paper.

### 4.2  $\delta$-MAPS for CERA-20C on 1901–2010

#### 4.2.1  Domain identification

Our algorithm, presented in Sect. 3.2.1, combines grid cells with highly rank correlated time evolution into domains. Domains have to be contiguous, but may overlap, grid cells may remain unassigned. Average mutual rank correlation within a domain 395  has to be higher than a selected threshold $\delta$, we examined the quantiles $q_{0.9}(\varrho_{ij}|i \neq j = 1, n) \leq \delta \leq q_{0.99}(\varrho_{ij}|i \neq j = 1, n)$ of all pairwise rank correlations. The plots included in this paper refer to thresholds $q_{0.95}$ for SST, and $\delta = q_{0.93}$ for Z500, chosen for their intuitive parcellation of the fields evocative of known teleconnection patterns. As varying the threshold effects the networks for different data sets in a similar way, the choice of $\delta$ changes the results only marginally.

    The domains constructed this way from the detrended, deseasonalized SST anomalies of the CERA-20C ensemble mean 400  include all important SST teleconnection patterns with interannual to decadal time scales (see for example Messié and Chavez (2011)). The map of the domains (Figure 1 (a)) resembles the corresponding maps for COBEv2 and HadISST in Falasca et al. (2019) reasonably well, taking into account the differing data sets and time periods. Their main domains are clearly identifiable: El Niño Southern Oscillation (ENSO, o11, for its broad extension also reminiscent to the region 2 of the Interdecadal Pacific Oscillation [IPO] tripol in Henley et al. (2015)); the Horse Shoe Pattern (o7); the South Pacific (o9); the Indian Ocean (o3); 405  the North Tropical Atlantic (o15, with extension to the extratropics); the South Tropical Atlantic (o1). Furthermore there are domains in the extra-tropical southern (o2) and eastern Indian Ocean (o4), the extra-tropical southern (o12) and north-eastern (o14) Atlantic and the sub-polar North Atlantic (o16), the Gulf Stream (o13), the North Pacific current (o8, region 1 of the IPO tripole), a domain corresponding to region 3 of the IPO tripole (o10), the Kuroshio Extension (o5), and a domain south of Australia (o6). Areas, where sea ice occurs, are omitted, because of the confounding effect on SST.

410     In the Z500 map of domains (Figure 1 (b)), the seasonally migrating Tropical Belt (TB, a15) formed by the Hadley circulation and the two polar cells (Arctic a1/a13 largely overlapping, and Antarctic a3) stand out, stretching around the whole globe. The mid-latitudes are populated by numerous domains with more (over ocean) or less (over land) pronounced zonal extension





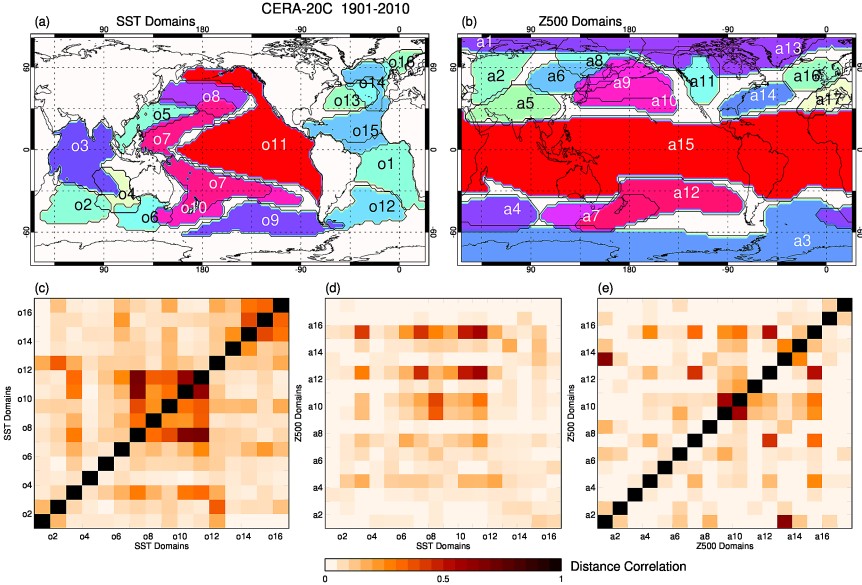

**Figure 1.** Domains of the CERA-20C ensemble mean (a) SST and (b) Z500 fields over the time period 1901–2010 (arbitrary colors). Maximum lagged distance correlation links between (c) SST and (e) Z500 domains and (d) cross-links.

(cyclone tracks). The missing segmentation of the tropical belt into several domains probably results from the seasonal time resolution.

### 4.2.2 Network of domains

The domains of SST and Z500 are now ready for network construction (see Sect. 3.2.2): Figure 1 (c) illustrates the distance correlations (Sect. 3.3) for all pairs of SST domains in the CERA-20C ensemble mean, omitting the geographic information for enhanced clarity. Only significant links to the FDR level $\alpha = 0.05$ (Sect. 3.2.2) are shown. However, even weak links are assessed significant, because the time series are long enough (4 seasons $\times$ 110 years) to allow the distance correlation to be estimated accurately. The darkest shades (except of the self links) correspond to the links between ENSO (o11), the Horse Shoe (o7) and IPO3 (o10): o11↔o7, o11↔o10, o7↔o10. We see enhanced connectivity of o7, o10 and o11 to the northern and southern Pacific Ocean (o8, o9) and from the Pacific to the Indian Ocean (o7/o10/o11↔o3), corresponding to known ENSO-teleconnections, but not to the Kuroshio Extension extension (o5). The southern Indian Ocean domain is furthermore linked to the South Atlantic (o2↔o12).

The intra-Atlantic links are much weaker: o14, o15 and o16 are largely overlapping domains and together supposedly form the Atlantic Multidecadal Oscillation (AMO); the Gulf Stream is linked to the north-eastern Atlantic (o13↔o14) as well as the tropical to the extra-tropical South Atlantic (o1↔o12). The South Atlantic is also weakly connected to all North Atlantic domains (o12↔o13/o14/o15/o16), but there is no link between the North Atlantic and the South Tropical Atlantic (o1). It might



be hypothesized that this bypass is related to the thermohaline circulation that tunnels the shallow subtropical cell (Liu and
Alexander, 2007). According to the network, the Atlantic is connected to the other oceans only via the Southern Ocean, with
links o12/o13/o14/o16↔o9, o14/o15/o16↔o6, o1/o12↔o2, and o16↔o2 that appear rather weak, although visible against
their virtually zero background. A link between the South Tropical Atlantic (o1) and ENSO (o11) as proposed in Falasca et al.
(2019) and Rodríguez-Fonseca et al. (2009) is not apparent in our network. This absence is likely caused by the non-stationarity
of this link, which was not observed before 1970. Nevertheless, it does appear, when a network is constructed for the period
1971–2010 (not shown).

Note that allowing for lagged dependence changes the network only marginally compared to a net with only instantaneous
links. Few connections are increased in strength of distance correlation by more than 0.05, and none by more than 0.1. All
links already exist in the instantaneous network, the structure of the network remains unchanged.

The network between Z500 domains (Figure 1 (e)) is considerably weaker than the SST network, possibly a consequence
of the stronger high-frequency variability of the Z500 time series in response to seasonally varying solar forcing combined
with weaker low-frequency variability caused by stronger mixing of the freely flowing air masses. Moreover, many of the
known atmospheric teleconnections vary considerably throughout the year, which weakens the all-season dependence between
the involved domains. Apart from the overlapping domains a1/a13, a9/a10, and a7/a12, the Tropical Belt (a15) is the most
strongly connected domain with links to the mid-latitudinal Ferrel cell domains, enveloping the cyclone tracks, over all oceans
(a15↔a4/a7/a9/a10/a12/a14), to which the undisturbed Hadley circulation releases a substantial amount of energy. Domains
over land have fewer and weaker links. Known atmospheric teleconnections are clearly identifiable: the Pacific North America
Pattern (PNA) with links a10↔a11, a10↔a14, but interestingly not a11↔a14, and the North Atlantic Oscillation (NAO)
with a link a13↔a14 (and much weaker a1↔a14). Other complex teleconnections also seem to involve the Arctic domain:
a1↔a2/a6/a8/a16 and a13↔a8. In contrast, the Antarctic domain (a3), is largely autonomous as discussed in Spensberger et al.
(2020). Lagged dependence is irrelevant in the Z500 network.

We notice that many known atmospheric teleconnections are defined as higher-order modes of some EOF decomposition.
As such they exist only as additive modulations of their corresponding leading modes. We would therefore not expect to find
many of them in our networks.

Network methods allow the investigation of interactions between different climatological fields in a straight forward way,
constructing cross-networks between (in our case) SST and Z500 domains that describe the coupled ocean–atmosphere vari-
ability (Liu and Alexander, 2007). We notice that the inference of links between the domains of two unipartite networks is
different from the construction of bipartite communities in multi-layer networks as in Ekhtiari et al. (2021). Here, we just cal-
culate the distance correlations between pairs of one SST and one Z500 domain. The inferred cross-links are shown in Figure 1
(d). The connectivity is mostly quite weak, except for the cross-links from the Tropical Belt (a15) and the northern and southern
Pacific Z500 domains (a9, a10, a12) to the ENSO-related SST domains (o7, o8, o9, o10, o11), the tropical Indian Ocean (o3),
but also the SST domain south of Australia (o6). This feature was also observed by Feng et al. (2012), who related it to the
Walker circulation. Z500 domains a4 and a7 participate in this pattern, but to a lesser extent. Z500 domains over oceans are
usually connected to their underlying SST counterparts (a4↔o2, a7↔o6, a9/a10↔o8, a14↔o13 [SST modulating the NAO],




a15↔o3/o11), although in the Atlantic this dependence is exceptionally weak (a15↔o1/o15, a16↔o14, a3↔o12). But tele-
connections to more distant SST domains are, in some instances, as strong as or even stronger than those proximate cross-links
(a4↔o3/o10/o11/o12, a7↔o12, a14↔o15). Interestingly, the Arctic Z500 domain (a13) is weakly linked to the AMO-domain
(o15), but not to the North Pacific. Except for slightly increased overall connectivity levels, supposedly mediated by the SST,
allowing for lagged dependence does not change the network.

### 4.2.3   3rd-order interactions

The overall high level of connectivity between SST domains motivated us to take a deeper look into the dependence structure
of the climate system. In a modest first attempt, we search for interacting triples in the sense of Lancaster, in graph theory
termed as 2-hyperedges, taking all combinations of three SST domains and calculating their distance 3rd-order multicorrelation
as introduced in Sect. 3.3.1 Equation 7. As discussed there, we choose a large FDR level $\alpha = 0.2$ in order not to suppress too
many distance multicorrelations. To avoid cumbersome evaluations with different lag combinations, we stick to instantaneous
networks.

Only a small number (13) of significant 3rd-order dependencies is detected (we list them in Table S1 instead of plotting
them), all somehow related to the ENSO phenomenon, one of them the IPO tripole. The hyperedges also include the tropical
Indian Ocean (o3) and the domain south of Australia (o6). As the nature of Lancaster Interaction is inherently non-linear, this
concentration on ENSO corresponds to the findings in Hlinka et al. (2014), who detect substantial non-linear contributions
to mutual information in SST (apart from trends and seasonal variance) mainly in the central tropical Pacific. Likewise, Pires
and Hannachi (2017) find synchronized extremes of uncorrelated PCs of SST in the Pacific that cannot be explained by linear
interaction. Even though, one distance multicorrelation is also detected in the North Atlantic: the SST triple (o14,o15,o16),
which corresponds to the AMO.

Note that not every triple with strong pairwise dependencies also has a significant 3rd-order dependence. Table S1 shows
the hyperedges along with their distance multicorrelation and the sum of their pairwise distance correlations. As distance
multicorrelation is symmetric, every significant hyperedge is listed only once in the table. Note also that the sum of pairwise
distance correlations is not bounded by 1, because the pairwise dependencies are not mutually exclusive. Although the detected
distance multicorrelations are significant, they are at most 20% of the sum of the respective pairwise distance correlations. That
means 3rd-order interactions complement, but not outweigh pairwise dependence in the 3-dimensional joint dependence.

The same comments essentially apply to cross-hyperedges consisting of two SST and one Z500 domains or one SST and
two Z500 domains. We detected 15 and 5 significant cross-hyperedges, respectively, in the Pacific, which all resemble some
ENSO interaction. The Z500 domains a13 and a14 (NAO) have no notable distance multicorrelation with North Atlantic
SST domains, indicating that the North Altlantic is linked to the NAO-domains on a pairwise basis (o15↔a13/a14), but no
higher-order interaction is taking place. There is no hyperedge of three Z500 domains with significant multicorrelation. Known
atmospheric tripoles like the Arctic Oscillation (a9, a13, a14) and the Pacific North America Pattern (a10, a11, a14) apparently
lack significant 3rd-order dependence.



We believe that the construction of higher-order networks including hyperedges by means of distance multicorrelation might well be one step towards understanding the synergies emerging from multivariate coupling of large-scale oceanic/atmospheric teleconnections.

## 4.3 Comparison of networks

### 4.3.1 Reference networks

We turn to the comparison of reference networks in terms of the NQS and MNQS criteria (see Sect. 3.4), calculated from the adjacency matrices $\mathbf{M}$ containing the regionally distributed distance correlation links between all pairs of domains (Equation 8). As CERA-20C was produced as a 10-member ensemble representing the inevitable sampling and modelling uncertainty inherent in the production process, we take this opportunity to construct the $\delta$-MAPS networks individually for each member. The results are matched to the networks derived for the CERA-20C ensemble mean.

The CERA-20C individual networks for the complete time period 1901–2010 are very similar to each other, with average NQSs close to 1 for all three parameters (average $NQS_{SST}= 0.98$, average $NQS_{Z500}= 0.94$, average $NQS_{SST–Z500}= 0.96$), such that the MNQSs have mean 0.96 with only a small spread. The average MNQS between the individual CERA-20C runs and the CERA-20C ensemble mean is 0.96. The small differences are brought about by the pattern correlation factor in netSSIM, the mean and variance factor being virtually equal to 1.

The networks for the shorter periods 1901–1955 and 1951–2010 are equally similar with average $MNQS= 0.95/0.95$ between runs and $0.95/0.96$ to the ensemble mean, respectively. Because the networks for individual CERA-20C runs and the CERA-20C ensemble mean are nearly indistinguishable, we will only take the CERA-20C ensemble mean networks for reference in the following comparisons.

When analysing the temporal evolution of the connectivity in the CERA-20C ensemble mean, we find good agreement between the first and second half of the century ($MNQS= 0.87$, Table 2), resulting from comparable differences in the SST and SST–Z500 networks, and higher similarity in Z500 ($NQS_{SST}= 0.84$, $NQS_{Z500}= 0.93$, $NQS_{SST–Z500}= 0.84$). In contrast, the full period is more similar to the first half in all networks ($MNQ= 0.96$, $NQS_{SST}= 0.95$, $NQS_{Z500}= 0.96$, $NQS_{SST–Z500}= 0.95$), than to the second half, because especially the SST–Z500 networks bear more differences ($MNQS= 0.92$, $NQS_{SST}= 0.92$, $NQS_{Z500}= 0.94$, $NQS_{SST–Z500}= 0.89$). We emphasize that the networks contain only information about the strength of the dependencies between the domains and not about their functional form.

Because of deviating domain extension and numbering, comparing the networks by means of the rectangular network plots (like in Figure 1 (c)–(e)) is cumbersome. In Figures S4–S7 we have plotted 2-modal slices of the spatially distributed adjacency hyper-matrices $\mathbf{M}$ wrt. grid cells in the ENSO-domain, in the AMO-domain and in the Tropical Belt, respectively. The comparison of these slices is evidently not exhaustive, but may give a hint on the nature of the differences between the networks.

The domains in the three CERA-20C SST network slices for the ENSO-domain (Figure S4 (a),(c),(e)) are very similar in shape and size, but the links between the domains are differently distributed. The networks most obviously disagree in link



**Table 2.** Multivariate Network Quality Scores between reanalyses in various time periods

|  | CERA-20C | | | 20CRv3 | | |
|---|---|---|---|---|---|---|
|  | 1901–1955 | 1951–2010 | 1901–2010 | 1901–1955 | 1951–2010 | 1901-2010 |
| CERA-20C | | | | | | |
| 1901–1955 | | 0.87 | 0.96 | 0.88 | | |
| 1951–2010 | 0.87 | | 0.92 | | 0.89 | |
| 1901–2010 | 0.96 | 0.92 | | | | 0.82 |
| 20CRv3 | | | | | | |
| 1901–1955 | 0.88 | | | | 0.81 | 0.84 |
| 1951–2010 | | 0.89 | | 0.81 | | 0.90 |
| 1901–2010 | | | 0.82 | 0.84 | 0.90 | |

strength from ENSO to the tropical Indian Ocean, but also from ENSO to the North Tropical Atlantic, to the North Pacific, and to the Southern Ocean. The same is visible in the network slices for the AMO-domain (Figure S5 (a),(c),(e)).

In contrast, the CERA-20C Z500 network slices for the Tropical Belt (Figure S6 (a),(c),(e)) bear more apparent similarity than the SST network slices, which was already apparent in the Network Quality Scores above. Although the shape of the Tropical Belt differs slightly more than the shape of the ENSO-domain, the links to the rest of the globe resemble each other

more strongly. However, the region over the North Pacific and the Antarctic domain seem somewhat ambiguous.

The cross-links from ENSO to the Z500 domains (Figure S7 (a),(c),(e)) and from the Tropical Belt to the SST domains (not shown) show similar differences like the unipartite networks. Yet, the stabilizing effect of the self-links (large patches with distance correlation 1) does not apply to the SST–Z500 cross-networks, such that the network scores may turn out a little lower.

As regards the second reanalysis 20CRv3, we observe strong similarity to the CERA-20C ensemble mean on the two shorter

time periods 1951–2010 and 1901–1955 (MNQS= 0.89 and MNQS= 0.88, Table 2 and Figures S4–S7 (a),(b) and (e),(f)), where disagreement within the same time period is mainly restricted to higher southern latitudes (remember that the SSIM includes an area weighting). But dissimilarities between the first and the second half of the century are stronger in 20CRv3 than in CERA-20C (MNQS= 0.81 and MNQS= 0.87, Table 2). Notably, in 1901–1955 20CRv3 shows the same strong connection between SST domains around the whole tropics as CERA-20C, which is lost in 1951–2010 in both reanalyses. In contrast,

the similarity between 20CRv3 and CERA-20C is slightly reduced in 1901–2010 (MNQS= 0.82, Table 2 and Figures S4–S7 (c)-(d)) mainly due to differing atmospheric interactions and the weaker cross-links in 20CRv3 compared to CERA-20C ($NQS_{SST}$= 0.94, $NQS_{Z500}$= 0.81, $NQS_{SST–Z500}$= 0.72). In all three networks (SST, Z500, SST–Z500) we observe that regional unsimilarity increases with latitude. Table S2 shows pairs of most similar domains between CERA-20C and 20CRv3 along with their domain-wise network quality score.

Besides, the similarity between the different time periods in 20CRv3 is not the same as in CERA-20C, with 1901–2010 more similar to 1951–2010 than to 1901–1955 (MNQS= 0.84 and MNQS= 0.90, Table 2 and Figures S4–S7 (b),(d),(f)). In contrast to CERA-20C, the link between ENSO and the South Pacific vanishes after 1950 in 20CRv3. This might be a consequence





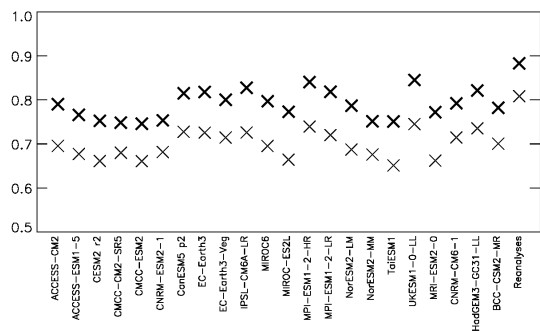

**Figure 2.** Average of Multivariate Network Quality Scores calculated using overall SSIM (bold) and point-wise SSIM (thin) of CMIP6 models over the time period 1951–2010 wrt. CERA-20C and 20CRv3.

of sparse observations in the first half of the century and thus a stronger dynamical heritage from the models used to produce the reanalyses. On the other hand, there might have been changes in connectivity driven by increasing GHG levels, which are

not equally reflected in CERA-20C and 20CRv3 (they are model results after all). Caution leads us therefore to restrict the comparison of CMIP6 data sets to reanalyses to the period 1951–2010.

### 4.3.2 CMIP6 networks

The networks belonging to the CMIP6 historical projections (listed in Table 1) are compared to the CERA-20C ensemble mean and the 20CRv3 best estimate on the time periods 1951–2010 in terms of individual NQSs, MNQSs and average MNQS in

Figures S8 and 2. As expected, the similarity between models and references is generally weaker than between references, although in the Z500 networks some models reach a comparable level. Network quality scores are highest for Z500, followed by SST and SST–Z500. SST–Z500 cross-networks show the greatest deviations across models as well as across references. The seemingly contradictory scores for Z500 wrt. CERA-20C and 20CRv3 have to be put into perspective with their very high values and can be traced back to the differences between the references.

When applying the alternative, point-wise SSIM calculation, the final average MNQS values are somewhat lower in their overall level, but similar in spread, and the model ranking suffers only minor changes (Figure 2).

The differences between the reanalyses are also reflected in the MNQSs of the models, where the reanalyses agree very well upon some models (HadGEM3-GC31-LL, IPSL-CM6A-LR, MPI-ESM11-2-HR, MIROC-ES2L, MIROC6), but less upon others (MRI-ESM2-0, TaiESM1, CNRM-CM6-1, CNRM-ESM2-1). But altogether, a tendency to differentiate between more/less

similar models wrt. reanayses is clearly visible. We conclude that, when combining several references from independent sources, the average MNQS over these references is a valid evaluation instrument for assessing, whether the teleconnections between large climate components in a general circulation model are realistically represented. Still, as our evaluation is restricted to a single run per model, we are not able to differentiate between good runs and good models as such.





Using the example of four of the highest-ranking GCM runs wrt. MNQS, we illustrate in short the opportunities offered
by the $\delta$-MAPS approach to detect model deficiencies. We examine some of the point-wise adjacency maps of EC-Earth3,
UKESM1-0-LL, MPI-ESM1-2-HR and IPSL-CM6A-LR in comparison to CERA-20C and 20CRv3 over 1951–2010 (Figures
S9–S13). In the SST networks, we notice that differences are not restricted to higher latitudes, as was the case for the two
reanalyses. Even in the main feature of interannual variability, ENSO, spatial connectivity deviates significantly. In all models
the tropical Indian Ocean depends much more strongly, although to varying degrees, on ENSO than in both reanalyses (Figure
S9). EC-Earth3 and IPSL-CM6A-LL not at all reproduce the northern extension of the ENSO domain seen in both reanalyses
(Figure S9 (a), (d), (e) and (f)), which reflects the widely recognized low-frequency interdependency between ENSO and the
Pacific Decadal Oscillation (PDO) (Henley et al., 2015). The links to the southern Indian Ocean and the South Atlantic differ
considerably across models, but no model shows better performance in all domains. In MPI-ESM1-2-HR the dependence
between AMO and ENSO is exaggerated, whereas in IPSL-CM6A-LR the Subpolar North Atlantic is nearly disconnected
from the Tropical North Atlantic, which is not consistent with AMO (Figure S10 (c) and (d)). As regards Z500, UKESM1-
0-LL shows an unrealistic link between the Tropical Belt and the Antarctic domain (Figure S11 (b)). At the same time, the
dependence of the Arctic domain is matched well *only* in UKESM1-0-LL (Figure S12 (b)). In contrast, the cross-links from
ENSO to Z500 are well-represented in all four models (Figure S13).

Continuing the analysis of all point-wise adjacency maps, it would be possible to identify regions/climate phenomena of
higher and lower confidence in any model. An exercise that might be instructive for both modelling groups and downstream
users of climate projections.

## 5    Conclusion

In order to evaluate the physical plausibility of CMIP6 GCM output, we have constructed functional interaction networks
within and between the SST and Z500 multivariate time series of two reanalyses (CERA-20C and 20CRv3) and 22 GCM
output data sets using the $\delta$-MAPS procedure. In response to several theoretical challenges related to the nature of long term
climate data, a number of innovations were introduced into $\delta$-MAPS:

- Detrending with season-reliant trend-EOFs

- Network construction using distance correlation

- Distance multicorrelation for higher-order interactions

- Network comparison with the structural similarity index

- Construction of a multi-reference multivariate network quality score

First of all, the two reanalyses were compared to one another in considerable detail, including the temporal evolution of the
interactions in the course of the 20[th] century. It could not be excluded that inconsistencies between the first and second half of
the century arise at least partly from data uncertainty. The evaluation of CMIP6 model output against the references revealed a

very high general similarity of the atmospheric connectivity, though with gradual differences. Oceanic teleconnections are less accurately reflected and the model differences more pronounced. The strongest deviations are found in the cross-networks between Z500 and SST, which co-occur sometimes, but not always, with lower network quality scores in the unipartite networks. We combined the three network quality scores for each CMIP6 model on an equal basis, emphasizing the equivalent importance of all considered geophysical subsystems in the generation of the earth's climate. Taking into account the uncertainty inherent

in any reference, the average multivariate network quality score over several, preferably independent, references can surely be considered a suitable criterion to assess the similarity of physical interactions between climate components in a model to those in observations.

In addition, the proposed complex networks framework combined with the distance correlation measure offers many promising multivariate extensions of $\delta$-MAPS as, for example, node definition based on multivariate time series, consideration of

higher-order dependence, interactions on multiple time scales and time-evolving networks. Such comparisons could be very useful to investigate subtle differences between various reanalyses. Besides, the characterization of network evolution from past to future could add a new facet to the understanding of climate change.

*Code and data availability.*  The $\delta$-MAPS software can be obtained in https://github.com/FabriFalasca/delta-MAPS. The GCM data used

in this study is part of the World Climate Research Programme's (WCRP) 6th Coupled Model Intercomparison Project (CMIP6) open access data. It was accessed through the Earth System Grid Federation (ESGF, https://esgf-node.llnl.gov/search/cmip6/). CERA-20C data is available at https://www.ecmwf.int/en/forecasts/datasets/reanalysis-datasets/cera-20c. 20CRv3 data is available at https://psl.noaa.gov/data/gridded/data.20thC_ReanV3.html.

*Author contributions.*  CD developed the concept, processed the data, prepared the manuscript, and produced all figures; KW and AW contributed with in-depth discussions, interpretation and review.

*Competing interests.*  The contact author has declared that neither she nor her co-authors have any competing interests.



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
