# Peer review of "Evaluation of Global Teleconnections in CMIP6 Climate Projections using Complex Networks"

_EGUsphere, 2022_

## Referee Comment (RC1)

The authors provide a Complex Networks perspective on climate model evaluation. The analysis is based on the d-MAPS methodology. The d-MAPS method has been used in climate science as a powerful way to investigate local and non-local connectivity patterns in spatiotemporal climate fields. And such method has been adopted for climate model evaluation and to study climate regime shifts in paleoclimate simulations.

The authors did not just adopt the d-MAPS framework but nicely contributed to it by proposing, adding and considering different new metrics. The first goal of this paper is then to improve the d-MAPS framework and to broaden its applications. The new metrics considered can quantify nonlinear relationships and to investigate linkages among multivariate fields. Moreover, several metrics for network comparison are also proposed, adding to the metrics initially proposed by Falasca et al. (2019). At a second step, the authors adopt such newly proposed metrics to investigate sources of differences and biases in CMIP6 models.

I believe that the paper can be suitable for publication in ESD following some revisions. As it is right now, I find it difficult to follow. The tools proposed are very interesting and powerful and there is lots of potential in their application. Novelty of such approach for climate model evaluation should be discussed in depth. What are we learning in terms of CMIP6 ensemble? What are the new tools showing us? This should be better discussed starting from the abstract. Part of this may be improved already by being clearer and more concise in few sections (such as Section 4.3.2). I provide comments below, ordered by sections.

**- Abstract**

The novelty of this work is proposing and implementing a series of different tools and metrics in the d-Maps framework. This is powerful as it further broadens the tools available in the d-MAPS methodology. This should be clearly stated in the abstract, currently is not. Please, revise the abstract by specifying that this is not only an application of d-MAPS, but it is an actual contribution to the overall d-MAPS framework. In fact, some modifications, such as the Spearman's Rank correlation in the domain identification step may result in very different results from the usual Pearson correlation in case of strongly nonlinear associations. So, I suggest to (a) rephrase the abstract in terms of the true novelty of the paper and (b) add some sentences on the results obtained with such new tools. Right now, there is only one line...what are the main results in the context of model evaluation? Which are the best models in terms of their network connectivity? Where do the models tend to fail?

**- Introduction**

- o Line 38: El Niõ -> El Niño
- Line 46: I suggest adding the paper of Tantet and Dijkstra to the references https://esd.copernicus.org/articles/5/1/2014/esd-5-1-2014.pdf

**- Data**

- What is the temporal resolution? I don't see it written (apologies in case I missed it)
- The paper uses two reanalyses products: CERA20-C and 20CRv3.
  In the data section can we see that CERA20-C is abbreviation for Coupled Reanalysis for the 20th century. Maybe I missed it, but I do not see anywhere that 20CRv3 is abbreviation for NOAA-CIRES-DOE Twentieth Century Reanalysis version 3 (also is not obvious). Please add the "20CRv3" abbreviation.
- Line 88: was there a reason to remap the two fields to two different resolutions? Why not both at 2.25 degrees?

**- Methods**

**• Domain identification**

Line 184-191: there is no need in this case to adopt the scheme proposed by Falasca et al. (2020). That scheme has proven to be useful to identify abrupt (and non-abrupt) shifts in climate variability at paleo-scales, which is not the goal of the submitted paper.

**• Network of domains**

In Line 208 the author say that every possible lag is analyzed from -L to L. What is L? Is it the length of the time series? Please clarify.

**• 3.3 Distance covariance/correlation.**

In line 264: "The correction of autocorrelation is a rather unrobust statistical technique...statistically advantageous". Is there a reference for this claim? If yes, please add the reference. If not elaborate on why that is the case.

**• 3.3.1 Distance multivariate/multicorrelation**

This is a nice and interesting addition to the d-MAPS framework. Results are discussed both in the case of distance covariance and distance multivariate. It is my understanding that in the case of Distance covariance/correlation, the authors considered all possible lags; while in the distance multivariate/multicorrelation they only consider *instantaneous* connections. Is this correct? If this is the case, can the author please add this point in the discussion of the metrics. At the risk of being repetitive I think it is useful to underline that (a) results with Distance covariance/correlation are computed using lags and that (b) results with distance multivariate/multicorrelation are instaneous. Adding to this, is there a reason for this choice? I understand the choice, as the problem may become easily intractable when considering *lagged* higher order dependence. This seems to be said in line 474: I suggest to be clear on this point in Section 3.3.1.

I find this metric very interesting. I am wondering on what are possible some connections/relationships with the following "Tripoles" concept/metric proposed in KDD: https://dl.acm.org/doi/pdf/10.1145/3097983.3098099 And used in this study:

https://journals.ametsoc.org/view/journals/clim/30/1/jcli-d-15-0884.1.xml?tab\_body=pdf

**• 3.4 Comparison of networks with structural similarity...**

Line 313: "Firstly our links are undirectional because distance correlation is much less sensitive to temporal lag than Pearson correlation, such that there is no distinct temporal ordering".

I am confused about this sentence. My understanding was that the network is a *direct* graph, inferred by considering *lagged* relationships. In fact by looking at Figure 1(c-e) the caption confirms "Maximum lagged distance links between..."

However, in the adjacency matrix proposed links are *undirectional*. Why this choice? Why not setting the entries in the adjacency matrix as the maximum link? This is not clear and should be discussed. Links in climate networks can be lagged as they reflect *non-local* connectivity. Such linkages are driven by atmospheric/ocean phenomena such as (for example) Rossby or Kelvin Waves. Therefore, physically many of such links cannot be instantaneous. Moreover, why are *all possible links investigated* (see line 208) if the distance correlation is much less sensitive to lags?

Please discuss this point. To help the reader would it be possible to compute one adjacency matrix using lagged-connections (for example considering the maximum lag) and the other one using instantaneous connections and show robustness? Importantly I suggest to clearly explain the choice of looking at instantaneous links for this part of the analysis, while still looking at lagged links in other sections.

Equation (9). How are the 3 constants  $c_1 c_2$  and  $c_3$  chosen?

**- Results**

**Detrending with trend EOF**

The detrending procedure proposed here is interesting. However, I find it difficult to follow the discussion when the Figures are all in the Supplemental Information. I suggest moving Figures S1 and S2 in the main paper.

The authors are convincing here in the sense of using trend-EOF to remove a forced trend. How do these spatial patterns of trends (Fiure S1(b)) compare to the simple linear detrending done for each grid cell. It would be useful to show (in the Supplemental Info) maps of slopes fitted from a simple linear regression (as usually done in climate studies) and discuss difference. This could help in making a stronger case for the use of trend-EOF.

**• Domain Identification**

Line 401: "The map of the domains (Figure 1(a)) resembles the corresponding maps for COBEv2 and HadISST in Falasca et al. (2019) reasonably well, taking into account the different data sets and time period." The maps in the two studies are very similar, however the similarity metric chosen here is Spearm's Rank, allowing "for monotone, yet non-linear associations" (line 179).

I think this comparison can teach us something about the connectivity structure of such fields. A simple conclusion here is that for the temporal resolution and fields considered time series are linearly dependent and linear methods are enough to infer local connectivity (i.e., domains). Additionally, it shows that how the trend is removed makes little impact on the overall result. I suggest to add such a discussion in the text.

**Networks of domains**

Figure 1(c-e): these metrics encode the "*Maximum lagged distance correlation*". I believe this means that for each couple of domains distance correlations have been computed for all lags and that the maximum (and also significant?) one has been chosen? What is confusing is that, at least by visual inspection, this matrix seems symmetric. If the matrix shows *direct (lagged)* links, it should be asymmetric. Please, can you explain why is this matrix symmetric?

Line 432-435: the S. Tropical Atlantic leading the ENSO domain has been shown to be a time-dependent link in both Falasca et al. (2019) (Section 8, "Climate networks in time") and in Martin Rey paper (see https://link.springer.com/article/10.1007/s00382-014-2305-3). So the fact that is not picked by the algorithm in the period 1901-2010 is expected. In both the domain identification step and the network inference step please be sure to say that Figure 1 is for CERA-20C and that results for 20CR3 are reported in the Supplemental Information.

**Section 4.2.3: 3rd order interactions**

Results for this section are in a table in the supplemental info. Please, move this table in the main text. In the Table caption please add description of what is dMcor and dCor to make it easier to follow.

Are the interactions reported in Table S1 just the significant ones? If yes, please add it in the caption.

**Section 4.3.1**

This section describes results in Figures S4-S7. It is difficult to follow as results are in the SI. I suggest to at least add Figures S4 and S6 and S7 just for one of the two reanalyses and discuss the other reanalysis (plus AMO link in the SI). In this way you would be able to show maps of strengths in the two univariate cases plus the multivariate case. Also, it makes it easier to follow.

Figures S4-S7: are these networks direct graphs (i.e., did you include lags)?

**Section 4.3.2**

The authors lost me here: (a) under reanalyses in Figure 2, we have the comparison between the two reanalyses. What are the models compared to? Are they compared to the ensemble mean of *CERA20* or *20CRv3* best estimates? It's difficult to interpret any of these results as it is not clear what they are compared to.

Figures S8 and Figure 2 should be both in the main text (and both in the same Figure).

Figure S8: difficult to understand. I suggest to describe the NQS and pointwise NQS in the case of CERA-20C and leave the comparison with 20CRv3 in the Supplement Info. Falasca, F., Bracco, A., Nenes, A., and Fountalis, I.: Dimensionality reduction and network inference for climate data us- ing  $\delta$ -MAPS: Application to the CESM Large Ensemble sea surface temperature, J. Adv. Model. Earth Sy., 11, 1479–1515, https://doi.org/10.1029/2019MS001654, 2019.

Falasca, F., Crétat, J., and Braconnot, P. Bracco, A.: Spatiotemporal complexity and time-dependent networks in sea surface temperature

from mid- to late Holocene, Eur. Phys. J. Plus, 135(5), 392, https://doi.org/10.1140/epjp/s13360-020-00403-x, 2020.

---

## Author Comment (AC1)

First of all, we would like to thank Reviewer 1 for engaging in the reviewing process and for providing his extraordinarily valuable comments, which will help to improve our manuscript considerably. It was a great pleasure for us to discuss the manuscript with an expert reviewer as competent in the field as this one.

We will answer the comments, highlight in blue, in detail in the following. The changes in the manuscript will be highlighted by yellow colorboxes.

**Abstract**

The novelty of this work is proposing and implementing a series of different tools and metrics in the d-Maps framework. This is powerful as it further broadens the tools available in the d-MAPS methodology. This should be clearly stated in the abstract, currently is not. Please, revise the abstract by specifying that this is not only an application of d-MAPS, but it is an actual contribution to the overall d-MAPS framework. In fact, some modifications, such as the Spearman's Rank correlation in the domain identification step may result in very different results from the usual Pearson correlation in case of strongly nonlinear associations. So, I suggest to (a) rephrase the abstract in terms of the true novelty of the paper and (b) add some sentences on the results obtained with such new tools. Right now, there is only one line...what are the main results in the context of model evaluation? Which are the best models in terms of their network connectivity? Where do the models tend to fail?

Inserted in abstract line 9: While doing so, a number of technical tools and metrics, borrowed from different fields of data science, are implemented into the  $\delta$ -MAPS framework in order to overcome specific challenges posed by our target problem. Those are trend-EOFs, distance correlation and distance multicorrelation, and the Structural Similarity Index metric.

Previous line 13 (now 17) changed and adjusted: The networks derived from projection data are compared to those from reanalyses. Our results indicate that no single climate projection outperforms all others in every aspect of the evaluation. But there are indeed models, which tend to perform better/worse in many aspects. Differences in model performance are generally low within the geopotential height unipartite networks, but higher in sea surface temperature and most pronounced in the bipartite network representing the interaction between ocean and atmosphere.

**Introduction**

Line 38: El Niõ -> El Niño now line 44, changed

Line 46: I suggest adding the paper of Tantet and Dijkstra to the references https://esd.copernicus.org/articles/5/1/2014/esd-5-1-2014.pdf Now line 52, added

**Data**

What is the temporal resolution? I don't see it written (apologies in case I missed it) Previous line 90 (now 97) "The analysis is carried out for seasonal anomalies on the overlapping time period from 1901 to 2010."

The paper uses two reanalyses products: CERA20-C and 20CRv3. In the data section can we see that CERA20-C is abbreviation for Coupled Reanalysis for the 20th century. Maybe I missed it, but I do not see anywhere that 20CRv3 is abbreviation for NOAA-CIRES-DOE Twentieth Century Reanalysis version 3 (also is not obvious). Please add the "20CRv3" abbreviation. Previous line 73 (now 80) added

Line 88: was there a reason to remap the two fields to two different resolutions? Why not both at 2.25 degrees? We chose the slightly lower resolution for the Z500 field only for the reason of

computational time. The calculation of the pairwise correlation is quite time consuming, and more so the rank correlation, and O(#grid cells2). As the geopotential is a very smooth variable, we thought the lower resolution is acceptable.

**Methods**

**Domain identification**

Line 184-191: there is no need in this case to adopt the scheme proposed by Falasca et al. (2020). That scheme has proven to be useful to identify abrupt (and non-abrupt) shifts in climate variability at paleo-scales, which is not the goal of the submitted paper. Completely true. We added this paragraph (now lines 191-199) to draw the readers' attention to this ingenious idea.

**Network of domains**

In Line 208 the author say that every possible lag is analyzed from -L to L. What is L? Is it the length of the time series? Please clarify. See previous line 216 (now 223) L=10, added "maximum lagged (- $10 \le L \le 10$ )" in previous line 416 (now 424)

**Distance covariance/correlation**

In line 264: "The correction of autocorrelation is a rather unrobust statistical technique...statistically advantageous". Is there a reference for this claim? If yes, please add the reference. If not elaborate on why that is the case. The correction of the sample cross-correlation for autocorrelation (AC) involves the calculation of empirical ACs for all possible lags. The greater the lag, the lower of course the number of data available for estimation. If a cut is applied to the ACs, this cut has to be estimated, too. Furthermore, the results for the variance of the empirical cross-correlation are asymptotical in nature and do not establish actual thresholds for finite sample estimates. They assume certain properties of the time series, which might well not be fulfilled in our case. This poses a considerable uncertainty on the finite sample Bartlett's formula. See Nan Su, Robert Lun (2012): Multivariate versions of Bartlett's formula. <a href="https://doi.org/10.1016/j.jmva.2011.08.008">https://doi.org/10.1016/j.jmva.2011.08.008</a> We softened the statement a bit to read (now line 270-272): "The correction of autocorrelation involves the estimation of a rather large number of autocorrelation coefficients. This might add to statistical uncertainty and its expendebility is therefore statistically advantageous."

**Distance multivariate/multicorrelation**

This is a nice and interesting addition to the d-MAPS framework. Results are discussed both in the case of distance covariance and distance multivariate. It is my understanding that in the case of Distance covariance/correlation, the authors considered all possible lags; while in the distance multivariate/multicorrelation they only consider instantaneous connections. Is this correct? If this is the case, can the author please add this point in the discussion of the metrics. At the risk of being repetitive I think it is useful to underline that (a) results with Distance covariance/correlation are computed using lags and that (b) results with distance multivariate/multicorrelation are instaneous. It is correct that the distance covariances are computed using lag, but distance multivorrelations are instantaneous. The reason for that is on one hand our decision to follow the original  $\delta$ -MAPS procedure, which allows lags. On the other hand, allowing lags in the multicorrelation would be extremely tedious. Subsection 3.2.2 clearly states the use of lags≤10 for the construction of the network of domains. In subsection 4.2.2 we added "maximum lagged (-10≤L≤10)" in previous line 416 (now 424).

Adding to this, is there a reason for this choice? I understand the choice, as the problem may become easily intractable when considering lagged higher order dependence. This seems to be said in line 474: I suggest to be clear on this point in Section 3.3.1. Subsections 3.3 and 3.3.1 only treat the

distance (multi)correlation as such, regardless of network construction. New line 484 states "To avoid cumbersome evaluations with different lag combinations, we stick to instantaneous networks."

I find this metric very interesting. I am wondering on what are possible some connections/relationships with the following "Tripoles" concept/metric proposed in KDD: https://dl.acm.org/doi/pdf/10.1145/3097983.3098099 And used in this study: https://journals.ametsoc.org/view/journals/clim/30/1/jcli-d-15- 0884.1.xml?tab\_body=pdf It seems as if the authors of these studies are searching for similar phenomena, but using very different techniques. Most notably, their approach is strictly linear, such that it is not completely clear to us, if it could at all find high-order interaction, which is inherently nonlinear. They have even extended their approach to multipoles: Saurabh Agrawal, Michael Steinbach, Daniel Boley et al. (2020): Mining Novel Multivariate Relationships in Time Series Data Using Correlation Networks. DOI : 10.1109/TKDE.2019.2911681 It is very unfortunate that we cannot compare their finding with ours as their tripole regions are much smaller than ours and  $\delta$ -MAPS doesn't remotely identify them. Nevertheless, we will keep that in mind for further research.

**Comparison of networks with structural similarity...**

Line 313: "Firstly our links are undirectional because distance correlation is much less sensitive to temporal lag than Pearson correlation, such that there is no distinct temporal ordering". I am confused about this sentence. My understanding was that the network is a direct graph, inferred by considering lagged relationships. In fact by looking at Figure 1(c-e) the caption confirms "Maximum lagged distance links between..." However, in the adjacency matrix proposed links are undirectional. Why this choice? Why not setting the entries in the adjacency matrix as the maximum link? This is not clear and should be discussed. Links in climate networks can be lagged as they reflect non-local connectivity. Such linkages are driven by atmospheric/ocean phenomena such as (for example) Rossby or Kelvin Waves. Therefore, physically many of such links cannot be instantaneous. Moreover, why are all possible links investigated (see line 208) if the distance correlation is much less sensitive to lags? Please discuss this point. To help the reader would it be possible to compute one adjacency matrix using lagged-connections (for example considering the maximum lag) and the other one using instantaneous connections and show robustness? Importantly I suggest to clearly explain the choice of looking at instantaneous links for this part of the analysis, while still looking at lagged links in other sections. We apologize for the equivocal wording. Figure 1(c) does indeed show the maximum lagged distance correlation. But the differences between the instantaneous distance correlation to the maximum lagged distance correlations are minimal (see previous line 436-438 (now 444-446) "Note that allowing for lagged dependence changes the network only marginally compared to a net with only instantaneous links. Few connections are increased in strength of distance correlation by more than 0.05, and none by more than 0.1. All links already exist in the instantaneous network, the structure of the network remains unchanged." We therefore conclude that distance correlation is not well suited for distinguishing the direction of an interaction. We changed the paragraph to "Firstly, our links are undirectional, because distance correlation is much less sensitive to temporal lag than Pearson correlation. The distance correlation coefficients for lags -10≤L≤10 differ only marginally from the value for L=0. So although we do construct M using maximum lagged distance correlation, we do not venture to infer the direction of the interaction from it." (new line numbers 321-323)

Equation (9). How are the 3 constants c1 c2 and c3 chosen? Changed to c1=c2=c3=0.00001. The constants are much lower than the actual values for  $\mu$ ,  $\sigma$  and  $\rho$  both from reanalyses and projections. We conducted a series of sensitivity test in this regard. The chosen values do not impact the results.

**Results**

**Detrending with trend EOF**

The detrending procedure proposed here is interesting. However, I find it difficult to follow the discussion when the Figures are all in the Supplemental Information. I suggest moving Figures S1 and S2 in the main paper. **inserted**

The authors are convincing here in the sense of using trend-EOF to remove a forced trend. How do these spatial patterns of trends (Fiure S1(b)) compare to the simple linear detrending done for each grid cell. It would be useful to show (in the Supplemental Info) maps of slopes fitted from a simple linear regression (as usually done in climate studies) and discuss difference. This could help in making a stronger case for the use of trend-EOF. Reviewer 1 is absolutely right, the comparison with seasonal linear detrending makes an extremely strong case in favour of t-EOFs. Figure added to the supplement as Figure S1.

**Domain Identification**

Line 401: "The map of the domains (Figure 1(a)) resembles the corresponding maps for COBEv2 and HadISST in Falasca et al. (2019) reasonably well, taking into account the different data sets and time period." The maps in the two studies are very similar, however the similarity metric chosen here is Spearm's Rank, allowing "for monotone, yet non-linear associations" (line 179). I think this comparison can teach us something about the connectivity structure of such fields. A simple conclusion here is that for the temporal resolution and fields considered time series are linearly dependent and linear methods are enough to infer local connectivity (i.e., domains). Additionally, it shows that how the trend is removed makes little impact on the overall result. I suggest to add such a discussion in the text. It is true that the impact of Spearman's rank correlation is marginal. The domains of either Spearman and Pearson do resemble each other a lot. Spearman's domains look a little more compact and, importantly, they present significantly less overlaps. As there is no scientific reason to prefer one over the other, we chose Spearman. On the contrary, the method of detrending does have an impact. Inspired by the reviewer's comment, we conducted an experiment on the CERA-20C SST data (1901-2010), where we changed the detrending method to linear, keeping all other parameters identical. The resulting unipartite network has a pattern correlation to the t-EOF-detrended network of only 0.83, which is less than those between the individual CERA runs. Even if the main features of the network are able to make it through linear detrending, Figure S1 speaks for itself. There is no point it applying an obviously inappropriate technique, if an appropriate one is affordable.

**Networks of domains**

Figure 1(c-e): these metrics encode the "Maximum lagged distance correlation". I believe this means that for each couple of domains distance correlations have been computed for all lags and that the maximum (and also significant?) one has been chosen? What is confusing is that, at least by visual inspection, this matrix seems symmetric. If the matrix shows direct (lagged) links, it should be asymmetric. Please, can you explain why is this matrix symmetric? As explained above, the distance correlation seems to be insensitive to time lags. We therefore plotted the maximum lagged distance correlation in the matrix regardless of temporal direction. The matrix is indeed symmetric.

Line 432-435: the S. Tropical Atlantic leading the ENSO domain has been shown to be a timedependent link in both Falasca et al. (2019) (Section 8, "Climate networks in time") and in Martin Rey paper (see https://link.springer.com/article/10.1007/s00382-014-2305-3). So the fact that is not picked by the algorithm in the period 1901-2010 is expected. **Exactly**. In both the domain identification step and the network inference step please be sure to say that Figure 1 is for CERA-20C and that results for 20CR3 are reported in the Supplemental Information. Previous line 401 (now 409) added "...the map of the CERA-20C SST domains (Figure 3(a))..."; previous line 410 (now 418) added "In the CERA-20C Z500 map of domains (Figure 3(b))..."; in previous line 416 (now 424) we already put "Figure 3(c) illustrates the distance correlations (Sect. 3.3) for all pairs of SST domains in the CERA-20C...", previous line 439 (now 447) added CERA-20C; previous line 458 (now 566) added CERA-20C SST—Z500; new line 477 added "The analogous plot for 20CRv3 can be found in Figure S1."; previous line 470 (now 478) added "Again, this subsection presents only results for CERA-20C over the time period 1901--2010."

**Section 4.2.3: 3rd order interactions**

Results for this section are in a table in the supplemental info. Please, move this table in the main text. In the Table caption please add description of what is dMcor and dCor to make it easier to follow. **done**

Are the interactions reported in Table S1 just the significant ones? If yes, please add it in the caption. done

**Section 4.3.1**

This section describes results in Figures S4-S7. It is difficult to follow as results are in the SI. I suggest to at least add Figures S4 and S6 and S7 just for one of the two reanalyses and discuss the other reanalysis (plus AMO link in the SI). In this way you would be able to show maps of strengths in the two univariate cases plus the multivariate case. Also, it makes it easier to follow. Figures for CERA-20C put into the main text. Figures for 20CRv3 remain in SI.

Figures S4-S7: are these networks direct graphs (i.e., did you include lags)? Yes, the distance correlations are again taken at their maximum over  $-10 \le L \le 10$ .

**Section 4.3.2**

The authors lost me here: (a) under reanalyses in Figure 2, we have the comparison between the two reanalyses. What are the models compared to? Are they compared to the ensemble mean of CERA20 or 20CRv3 best estimates? It's difficult to interpret any of these results as it is not clear what they are compared to. Previous line 558-560 (now 566-570) changed to "The networks belonging to the CMIP6 historical projections (listed in Table 1) are compared in Figure 4 to the CERA-20C ensemble mean (bold black cross marks) and to the 20CRv3 best estimate (bold red cross marks) on the time period 1951--2010 in terms of individual network NQSs (for SST networks (a), for Z500 networks (b) and for the cross-networks (c)) and in terms of MNQSs for each reference respectively (d). Finally we take the average of both MNQSs to account for the uncertainty inherent in the reanalyses: 1/2(MNQS(CERA-20C)+MNQS(20CRv3)) ((e), bold crossmarks)."

Figures S8 and Figure 2 should be both in the main text (and both in the same Figure). done

Figure S8: difficult to understand. I suggest to describe the NQS and point wise NQS in the case of CERA-20C and leave the comparison with 20CRv3 in the Supplement Info. We hope the above description clarifies the issue.

---

## Author Response (AR1)

All comments of the editor have been answered, and all indications have been followed.

All comments of reviewer 1 and reviewer 2 have been answered in the respective replies. All requested changes and amendments have been inserted in the manuscript, they are marked-up in yellow in the marked-up manuscript.